# Therapeutic potential of fisetin in hepatic steatosis: Insights into autophagy pathway regulation and endoplasmic reticulum stress alleviation in high-fat diet-fed mice

Mahboobe Sattari[1], Mohammad Esmaeil Shahaboddin[2], Maryam Akhavan Taheri[3], Ehsan Khalili[1], Ozra Tabatabaei-Malazy[4, 5], Golnaz Goodarzi[6], Sadra Samavarchi Tehrani[7], Reza Meshkani[1]*, Ghodratollah Panahi[1]*

**1** Department of Clinical Biochemistry, Faculty of Medicine, Tehran University of Medical Sciences, Tehran, Iran, **2** Research Center for Biochemistry and Nutrition in Metabolic Diseases, Institute for Basic Sciences, Kashan University of Medical Sciences, Kashan, Iran, **3** Anatomical Sciences Research Center, Institute for Basic Sciences, Kashan University of Medical Sciences, Kashan, Iran, **4** Non-Communicable Diseases Research Center, Endocrinology and Metabolism Population Sciences Institute, Tehran University of Medical Sciences, Tehran, Iran, **5** Endocrinology and Metabolism Research Center, Endocrinology and Metabolism Clinical Sciences Institute, Tehran University of Medical Sciences, Tehran, Iran, **6** Department of Pathobiology and Laboratory Sciences, School of Medicine, North Khorasan University of Medical Sciences, Bojnurd, Iran, **7** Endocrine Research Center, Institute of Endocrinology and Metabolism, Iran University of Medical Science, Tehran, Iran

* rmeshkani@tums.ac.ir (RM); pshahriyar@gmail.com, ghpanahi@sina.tums.ac.ir (GP)

## Abstract

Non-alcoholic fatty liver disease (NAFLD) is a common condition with limited FDA-approved treatments due to its complex pathogenesis. Metabolic stress-induced lipotoxicity triggers the unfolded protein response, leading to the development of NAFLD through inflammation and apoptosis. Moreover, metabolic dysregulation compromises autophagic capacity, impairing effective ERphagy and lipophagy in the liver. Fisetin (FSN), a flavonoid present in various fruits and vegetables, has demonstrated the ability to regulate the processes mentioned above and possesses a range of biological properties. In this study using a high-fat diet-induced NAFLD mouse model, treatment with FSN at a dosage of 80 mg/kg per day for eight weeks resulted in reduced hepatic lipid accumulation. This effect was mediated by modulating ER stress through enhancing autophagic activity, as indicated by decreased expression of GRP78, elf2a, ATF4, and CHOP genes, along with increased AMPK phosphorylation, decreased mTOR expression, and elevated levels of ULK1, ATG5, and Beclin1. Additionally, there was an increase in the LCII/LC3I ratio and a reduction in p62 levels in hepatic tissue. Our findings suggest that FSN exerts its effects by activating the AMPK/mTOR signaling pathway and its downstream targets, underscoring its potential therapeutic advantages in managing NAFLD by targeting autophagy and ER stress pathways.

**Data availability statement:** All relevant data are within the paper and its Supporting Information files.

**Funding:** This work was supported by the Department of Clinical Biochemistry, Faculty of Medicine, Tehran University of Medical Sciences, Tehran, Iran in the form of a grant (1401-4-101-63977) received by GP. No additional external funding was received for this study. The funder had no role in study design, data collection and analysis, decision to publish, or preparation of the manuscript.

**Competing interests:** The authors have declared that no competing interests exist.

## Introduction

Non-alcoholic fatty liver disease (NAFLD), which is caused by hepatic lipid overload, is the most globally prevalent chronic liver disease. The intricate pathogenic mechanism has posed challenges in the development of drugs for NAFLD, as no related medications have been approved by FDA. As such, the first line of treatment for NAFLD remains lifestyle modification [1,2]. However, because long-term weight loss diets are particularly difficult to stick to [3], there is considerable interest in identifying agents affecting key pathogenic pathways in order to treat or prevent this illness.

NAFLD is often associated with obesity, insulin resistance (IR), and metabolic syndrome [4]. Free fatty acids (FFAs) are widely mobilized from adipocytes to the liver as a result of IR. This process increases the amount of β-oxidation of FFAs in the mitochondria, which causes electrons to leak out of the electron transport chain. This can result in mitochondrial dysfunction and oxidative stress [5]. At high concentrations, reactive oxygen species cause oxidative modifications to cellular compounds and causes a build-up of damaged macromolecules, resulting in liver injury [6]. In stress conditions like excessive FFA levels and lipotoxicity, improperly folded and unfolded proteins amass in the endoplasmic reticulum (ER) lumen, triggering the unfolded protein response (UPR) to reestablish ER equilibrium, as shown in NAFLD patients [7,8]. ER stress caused by metabolic stress is associated with changes in membrane and lipid biosynthesis, inflammation, and apoptosis, adding to the transition from simple steatosis to steatohepatitis [9]. Autophagy, as an adaptive mechanism in cellular damage, plays a crucial role in the removal and engulfment of damaged organelles and misfolded proteins [10]. Also, a specific type of autophagy, known as lipophagy, involves the breakdown of lipid droplets in the cytosol. It has been documented that NAFLD patients exhibit decreased autophagy [11,12]. The molecular mechanism underlying autophagy dysfunction in NAFLD involves both short-term regulations, where the pathway is inhibited by the mTOR complex activated due to overfeeding, and long-term regulation, where hyperinsulinemia contributes to its inhibition [13]. Furthermore, high-fat diet-induced changes in membrane lipid composition exacerbate the reduction of autophagy by causing defects in autophagosome-lysosome fusion [14]. The compromised autophagic capacity fails to effectively remove damaged organelles [15,16]. Since autophagy defects contribute to a vicious cycle in various aspects of NAFLD pathogenesis, the induction of autophagy may represent a promising target for therapeutic interventions [13,17].

Herbal compounds, particularly flavonoids, are gaining popularity as non-toxic and preventive medicinal treatments [18]. Flavonoids exhibit beneficial effects against NAFLD by preventing hepatosteatosis, reducing calorie intake, body weight, and visceral fat deposition, while acting as potent antioxidants to mitigate inflammation and insulin resistance [19]. Furthermore, in NAFLD patients, flavonoids improve blood pressure, glucose tolerance, insulin sensitivity, dyslipidemia, and adiponectin levels [20]. Fisetin (FSN), a flavonoid present in vegetables and fruits, has demonstrated numerous biological properties, including anti-invasive, anti-angiogenic, anti-inflammatory, antioxidant, and regulation of cell metabolism and autophagy in various conditions such as cancer, diabetes, cardiovascular and neurological diseases

[21,22]. In this study we investigate how FSN affects NAFLD by examining autophagy and ER stress, key processes linked to NAFLD advancement. To investigate the role of autophagy in NAFLD, we used hydroxychloroquine (HCQ) a lysosomotropic drug that inhibits autophagy and macroautophagy. HCQ prevents lysosomal degradation by increasing the pH within lysosomes and we aimed to understand how the suppression of autophagy affects the progression and pathology of NAFL [23]. The study's insights provide information about FSN's therapeutic potential for NAFLD and its complications, offering promise for the development of safe and effective treatments.

## Materials and methods

### Drugs and chemicals

FSN, with a (HPLC) purity of ≥ 98% and CAS No. 528-48-3 (Batch No. TF20220119), was acquired from Henan Tianfu Chemical Co. Ltd, located in Henan, China. HCQ with CAS No. 747-36-4 (Batch No. HCS0040299) was provided by Tehran Chemical Ph. Raw Material Co. in Tehran, Iran. The primary antibodies utilized in western blot analysis comprised LC3B (#2775 Cell Signaling), p62 (sc-10117, Santa Cruz, CA), p-AMPKα1/2 (sc-33524, Santa Cruz, CA), AMPKα1/2 (sc-74461, Santa Cruz, CA), and β-Actin (sc-517582, Santa Cruz, CA). The secondary antibody employed in western blot analysis were the mouse anti-rabbit IgG-HRP antibody (sc-2357, Santa Cruz, CA) and m-IgGκBP-HRP (sc-516102, Santa Cruz, CA).

### Animals

Male C57BL/6J mice weighing 18–20 g at 7–8 weeks of age were acquired from the Iranian Pasteur Institute. They were housed in a controlled environment with a constant temperature of 23–25°C, humidity ranging from 50–60%, and a standard 12-hour light/12-hour dark cycle. The mice had *ad libitum* access to water and pathogen-free food. Each cage was provided with enrichment (plastic tube and shredded paper). The cages were checked daily and cleaned once a week. All animal experiments were conducted according to guidelines approved by the Research Ethics Committees of Laboratory Animals at Tehran University of Medical Science (IR.TUMS.AEC.1401.126).

### Experimental protocol

Following a two-week adaptation period, one of two diets was given to mice at random: the standard chow diet (SCD) (10.2% fat and 71.5% carbohydrates; D12450J; Research Diets) (n = 8), and the high-fat diet (HFD) (60% calories from fat; D12492; Research Diets) (n = 40) to induce NAFLD model [24]. After 16 weeks, five groups were randomly selected from the HFD group (n = 8): HFD; HFD+Vehicle (V); HFD + FSN; HFD + HCQ and HFD + FSN + HCQ. FSN was dissolved in DMSO and tween80, then diluted in 0.9% normal saline for daily gavage administration to mice (80 mg/kg) alongside the HFD for 8 weeks. HCQ in buffered aqueous solution was injected intraperitoneally (i.p) twice a week (50 mg/kg, 0.1 ml/10g body weight) for 8 weeks (Fig 1a). Dosage adjustments were made according to the weekly body weight. Mice diet intake was tracked daily, and body weight was recorded every week. Caloric intake was calculated as the weight (g) of food consumed times the diet's calories (kcal) [25]. The %weight gain/ caloric intake (gkcal) x 100 formula is used to compute the food efficiency ratio [26].

### Metabolic tolerance tests

Metabolic tolerance was assessed at the end of the treatment period through oral glucose tolerance test (OGTT) and insulin tolerance test (ITT) [27]. For the OGTT, blood glucose levels were measured after a 10-hour nocturnal fasting period (from 8 PM to 6 AM). Subsequently, an α-D-glucose solution (10%) was administered (2 g/kg) by gavage, and blood glucose levels were monitored at 15, 30, 60, and 120-minute intervals. For the ITT, blood glucose levels were measured after a 6-hour morning fasting period (from 6 AM to 12 PM). Then, an insulin solution (0.25 IU/ml) was administered intraperitoneally (0.75 IU/kg), with blood glucose levels monitored at 15, 30, 60, and 90-minute intervals. The measurements utilized a glucometer (On Call Plus®) to analyze a blood drop from the mice's tail tip.

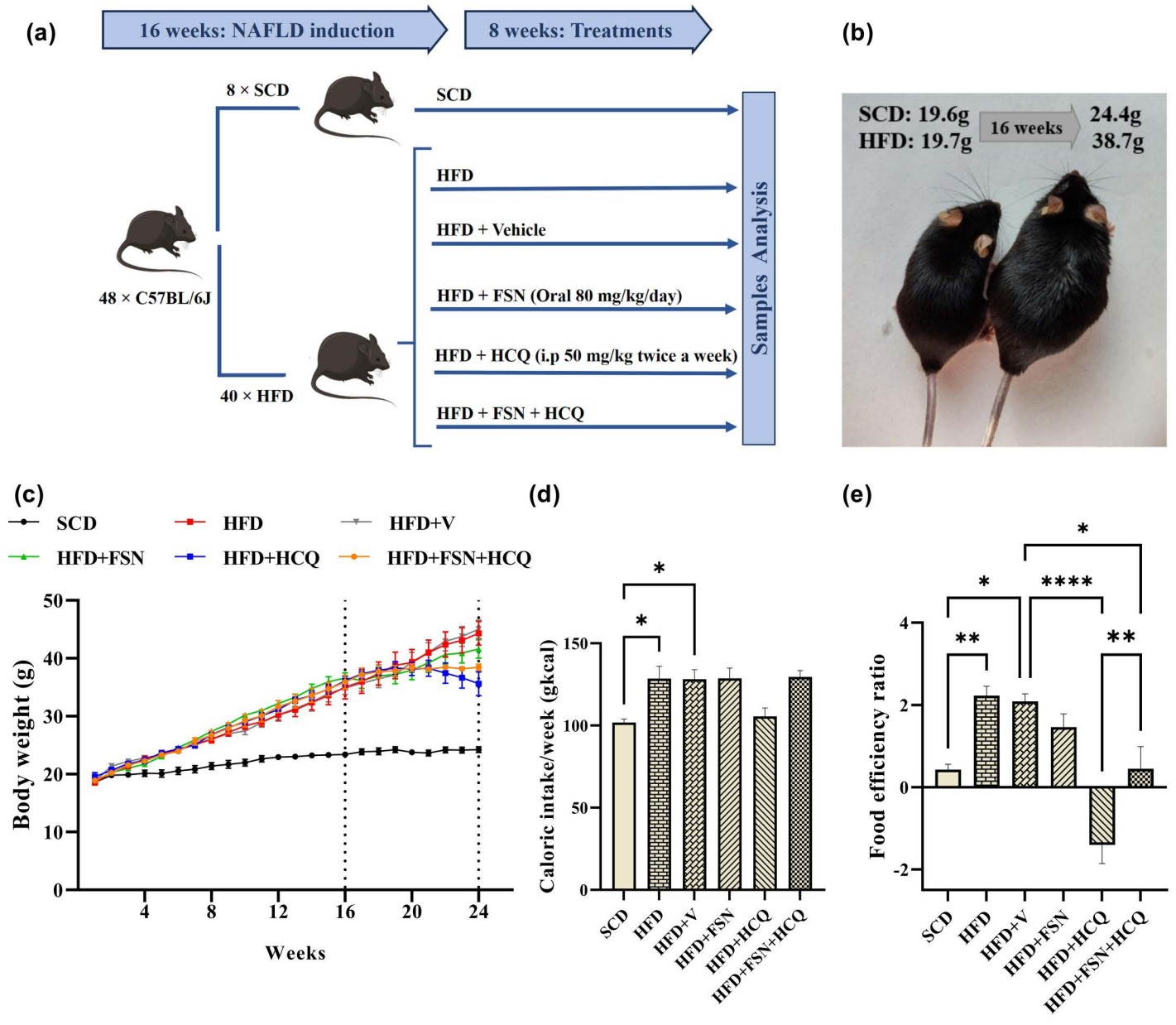

**Fig 1. Fisetin alleviates HFD-induced obesity and food efficiency ratio in mice. (a)** Schematic diagram of the experimental procedure. **(b)** A mouse fed SCD on the left and HFD on the right with their weight gain after 16 weeks. **(c)** Change of body weight during a 24-week study period. **(d)** The average Caloric intake and **(e)** food efficiency ratio during the 8th week of treatment (n = 8). The information is displayed as mean ± S.E.M. *P < 0.05, **P < 0.01, ***P < 0.001, and ****P < 0.0001.

## Sampling

The mice were allowed to fast for ten hours at the conclusion of the treatment period. Prior to euthanasia, deep anesthesia was induced using inhalation of $CO_2$. Following this, the pedal reflex was checked to confirm the depth of anesthesia. Once confirmed, rapid decapitation was performed using a guillotine. After being drawn into a tube, about 0.8 ml of blood was left to stand at room temperature for 20 minutes. Following a 10-minute centrifugation at 3000 rpm for the blood sample, different biochemical parameters were examined in the supernatant. Subsequently, the body was dissected, and the

liver tissues were removed, weighed, and divided into multiple portions for different experimental objectives. One part was put into a tube that had been filled with a formalin solution (%4). For subsequent analysis in Western blot and qRT-PCR studies, the remaining tissues were frozen in liquid nitrogen and placed in a refrigerator at −80°C.

## Biochemical parameters analysis

Serum levels of triglyceride (TG), total cholesterol (TC), LDL-C, HDL-C, alanine aminotransferase (AST) and aspartate aminotransferase (ALT) were determined with a kit that can be purchased commercially (Zeist Chem Diagnostics. Tehran, Iran) in CL2000i Autoanalyzer.

## Liver histological analysis

Liver tissue specimens were preserved for a minimum of twenty-four hours in paraformaldehyde. Following a graded series of ethanol dehydration, the specimens were cleaned in xylene and embedded in paraffin wax. Using a rotary microtome, tissue blocks were sectioned at a 5 µm thickness. Hematoxylin and eosin were used to stain the sections. Histological features were viewed using an Olympus light microscope (Olympus, Tokyo, Japan) by a qualified pathologist. The liver histopathological changes were evaluated by calculating the NAFLD activity score (NAS), which considered four criteria: steatosis, ballooned hepatocytes, lobular inflammation, and fibrosis stage [28]. To quantify fat vacuoles, randomly selected images were captured at 400X magnification from the pericentral hepatocytes region. The quantification was carried out using ImageJ software, focusing on two variables: particle count and percentage of the surface area.

## RNA extraction and quantitative real-time PCR

The Favorgen Tissue Total RNA Mini Kit (Ping Tung, Taiwan) was utilized to extract total RNA from fresh frozen liver tissues. The quality and concentration of extracted RNA were assessed using a NanoDrop™ device (Thermo Scientific NanoDrop OneC). A cDNA synthesis kit was used to reverse transcribe 1000 ng of extracted RNA into cDNA (yekta tajhiz azma, Tehran, Iran). The hepatic mRNA levels of key molecules involved in UPR regulation including Glucose-Regulated Protein 78 (GRP78), activating transcription factor 4 (ATF4), Eukaryotic initiation factor 2A (eIF2A) and C/EBP homologous protein (CHOP), as well as molecules involved in the regulation of autophagy, such as the mechanistic target of rapamycin kinase (mTOR), unc-51-like autophagy-activating kinases 1 (ULK1), ATG6/Beclin-1 (BECN1), autophagy related 5 (ATG5), and Sequestosome 1 (p62/SQSTM1) were measured using YTA® Universal qPCR Master Mix (Yekta Tajhiz Azma, Tehran, Iran) on a real-time PCR cycler (Qiagen Rotor-Gene Rt Pcr Machine). We used beta-actin as the housekeeping gene for normalization of qPCR data. The expression of beta-actin did not show any significant differences between the treated groups. Table 1 in Appendix S1 Table contains a list of the primers that were used in this investigation to measure the mRNA expression levels in Real-Time PCR.

## SDS-Page and western-blot analysis

Lysis buffer was used to homogenize mouse liver tissues. The Bradford method was used to measure the protein concentrations in the supernatant after centrifugation [29]. The reference protein was bovine serum albumin. Protein lysate in equal amounts was directly loaded onto a 10–12% SDS-PAGE and then placed onto a PVDF membrane for protein blotting. The membranes were incubated with the designated primary antibodies for 16 h at 4°C after being blocked for 75 min at room temperature using 2% non-fat dry milk (w/v) in Tris-buffered saline with 0.1% Tween 20 detergent (TBS-T). After that, membranes were probed for 75 min at room temperature using either m-IgGκBP-HRP or mouse anti-rabbit IgG-HRP secondary antibodies. Each gel lane contained an aliquot of the pooled standard sample. The ECL advanced reagents were used to identify the immuno-reactive bands. β-actin functioned as a loading regulator. With Image J, a densitometric analysis was carried out.

 

### Statistical analyses

The data is shown as mean±S.E.M. One-way ANOVA was used to identify statistical differences, and Tukey's multiple comparison test was then performed. P values less than 0.05 were deemed statistically significant. *P<0.05, **P<0.01, ***P<0.001, and ****P<0.0001 denote statistically significant differences. To conduct statistical analyses, Prism 9 (GraphPad, San Diego, CA) was used.

## Results

### Fisetin alleviates HFD-induced obesity and food efficiency ratio in mice

We fed HFD to mice for 16 weeks in order to induce NAFLD (Fig 1b). As depicted in Fig 1c, from week 4 until the end of the experiment, the HFD-fed mice's body weight increased significantly more than that of the SCD group. This was due to the augmented mean caloric intake in HFD group in comparison to SCD group. After 8 weeks of drug treatment, there were no variations found in the caloric intake analyzed between groups consuming HFD, except for the HCQ group (20%) (Fig 1d), which exhibited a significant weight loss compared to other groups. Despite the same caloric intake, FSN slightly and FSN+HCQ significantly hindered the food efficiency ratio in contrast to HFD mice (Fig 1e).

### Fisetin enhances glucose homeostasis in HFD-fed mice

Comparing HFD-fed mice to SCD-fed mice, the former had higher levels of fasting plasma glucose (FPG), both after 6 and 10 hours, while mice in treatment groups had a reduction that became significant for HCQ and FSN+HCQ groups in the longer time of fasting (Fig 2a, 2b). SCD mice were demonstrated to have a glucose tolerance by the OGTT, while mice fed on HFD appeared to have a decreased glucose tolerance. The glucose tolerance of mice treated with FSN and FSN+HCQ was significantly improved (Fig 2c). The total area under the curve (AUC) ascertained the above results (Fig 2d). During ITT, SCD and HFD reached under 50% of basal concentration of glucose about by 30 and 60 minutes, respectively after insulin injection. The treated groups showed this reduction in a time closer to the SCD group (Fig 2e). The AUC elucidated that insulin sensitivity improved statistically after FSN, HCQ and FSN+HCQ treatment (Fig 2f, 2g).

### Fisetin preserve lipid homeostasis in HFD-fed mice

Serum TG was significantly reduced by all three types of treatment compared to HFD (Fig 3a). FSN treatment didn't reduce serum TC (Fig 3b), but significantly reduced LDL-C (Fig 3c). Treatment with FSN+HCQ significantly reduced serum TC (Fig 3b), accompanied by a decrease in LDL-C (Fig 3c), while the proportion of HDL-C increased (Fig 3d).

### Fisetin alleviates NAFLD development in HFD-fed mice

Grossly, the HFD group showed fatty liver-like changes with hepatomegaly, paler coloration, and greasy appearance. In contrast, the normal group had dark brown/red livers, while the FSN and FSN+HCQ groups regained partly this color (Fig 4a). H&E staining indicated that treatment with FSN, HCQ and FSN+HCQ significantly alleviated NAS in the livers of mice induced by HFD (Fig 4a, 4b). This cumulative score was mainly due to the suppressed individual components of steatosis and inflammation (Fig 4c–4e). Moreover, the quantitative data based on lipid droplet count and their surface area percentages further supported the findings related to NAS (Fig 4f, 4g). Additionally, the HFD group was observed to have considerably increased liver weight/body weight (LW/BW) (Fig 4h), hepatic TG and TC contents (Fig 4i, 4j), as well as, serum liver enzymes (AST and ALT) (Fig 4k, 4l) compared with SCD groups. In comparison to the untreated group, these parameters were significantly lower in the FSN+HCQ-treated group. Although FSN did not have a significant effect on the relative weight of the liver, it improved all of other mentioned parameters. According to the findings, an 8-week FSN and FSN+HCQ treatment protected liver from lipid accumulation under HFD.

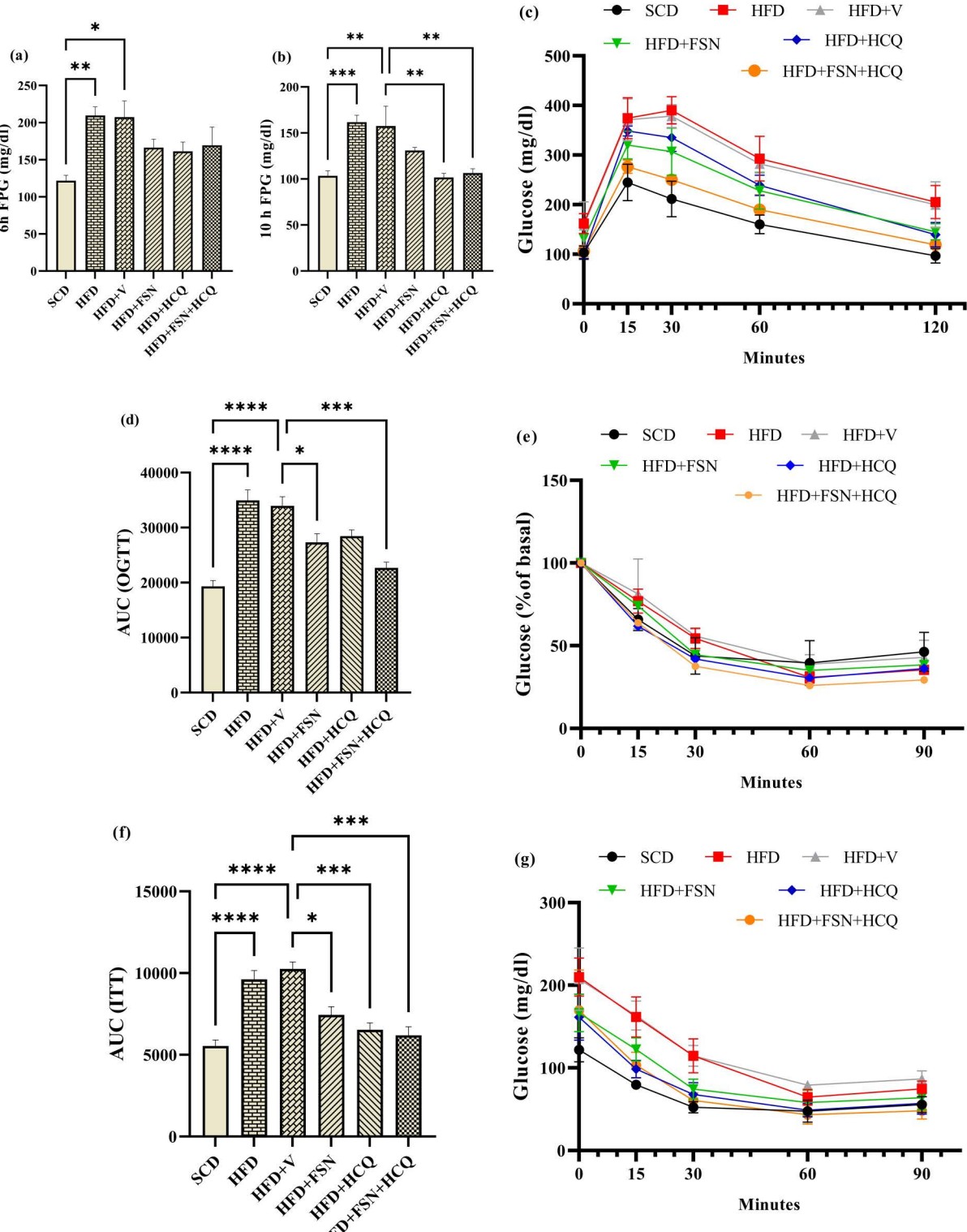

**Fig 2. Fisetin enhances glucose homeostasis in HFD-fed mice.** The average level of fasting plasma glucose (FPG) in mice after **(a)** 6 hours and **(b)** 10 hours of fasting. **(c)** Oral glucose tolerance test (OGTT) and **(d)** total area under the curve (AUC) of blood glucose level. **(e)** Mean percent reduction of blood glucose from baseline on time points after insulin injection. **(f)** Insulin tolerance test (ITT) and **(g)** AUC of blood glucose level (n = 5). The information is displayed as mean ± S.E.M. *P < 0.05, **P < 0.01, ***P < 0.001, and ****P < 0.0001.

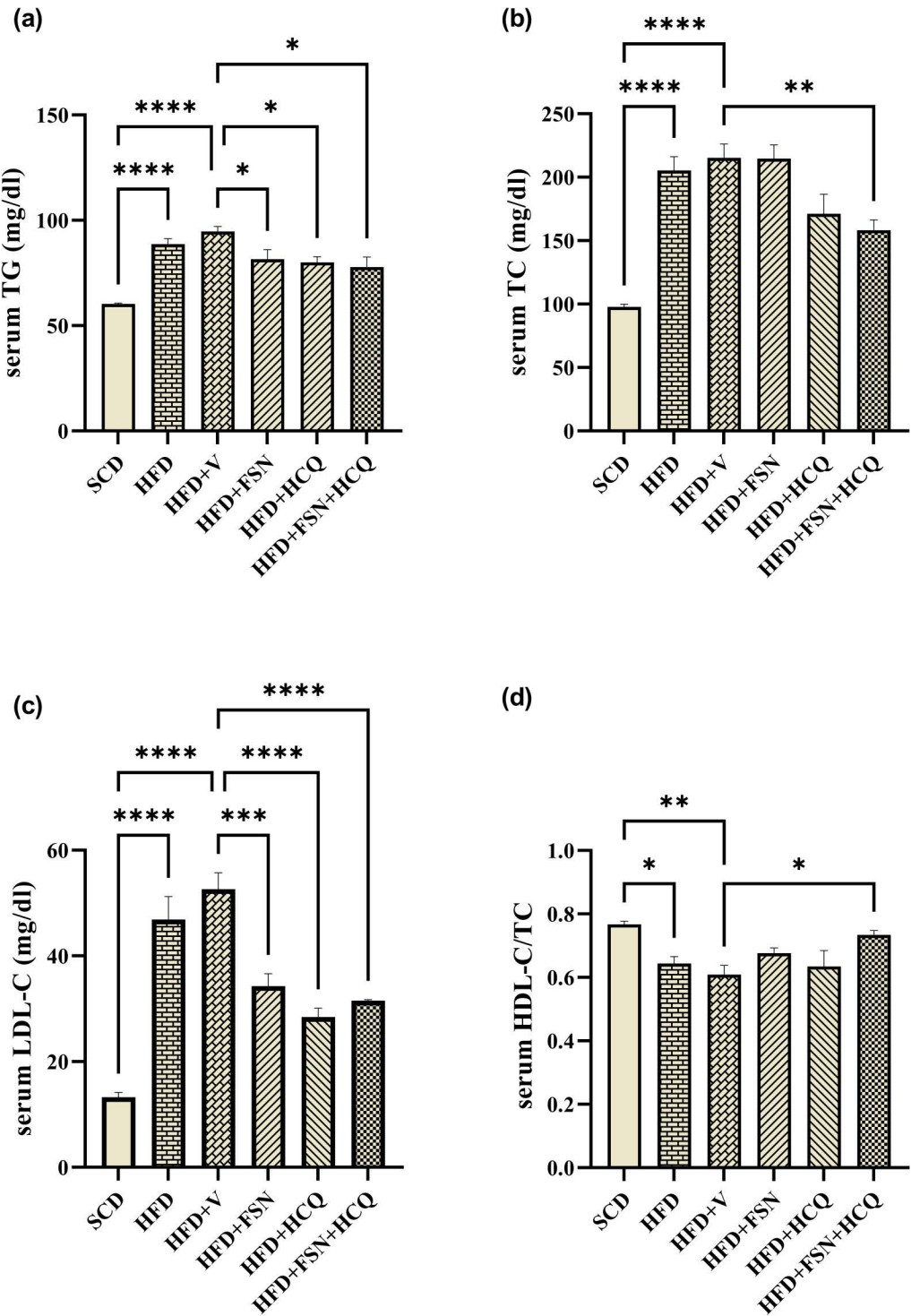

**Fig 3. Fisetin preserve lipid homeostasis in HFD-fed mice.** Serum levels of lipid biochemical parameters in mice. **(a)** Triglyceride (TG), **(b)** Total cholesterol (TC), **(c)** LDL-C and **(d)** HDL-C/TC (n=5). The information is displayed as mean±S.E.M. *P<0.05, **P<0.01, ***P<0.001, and ****P<0.0001.

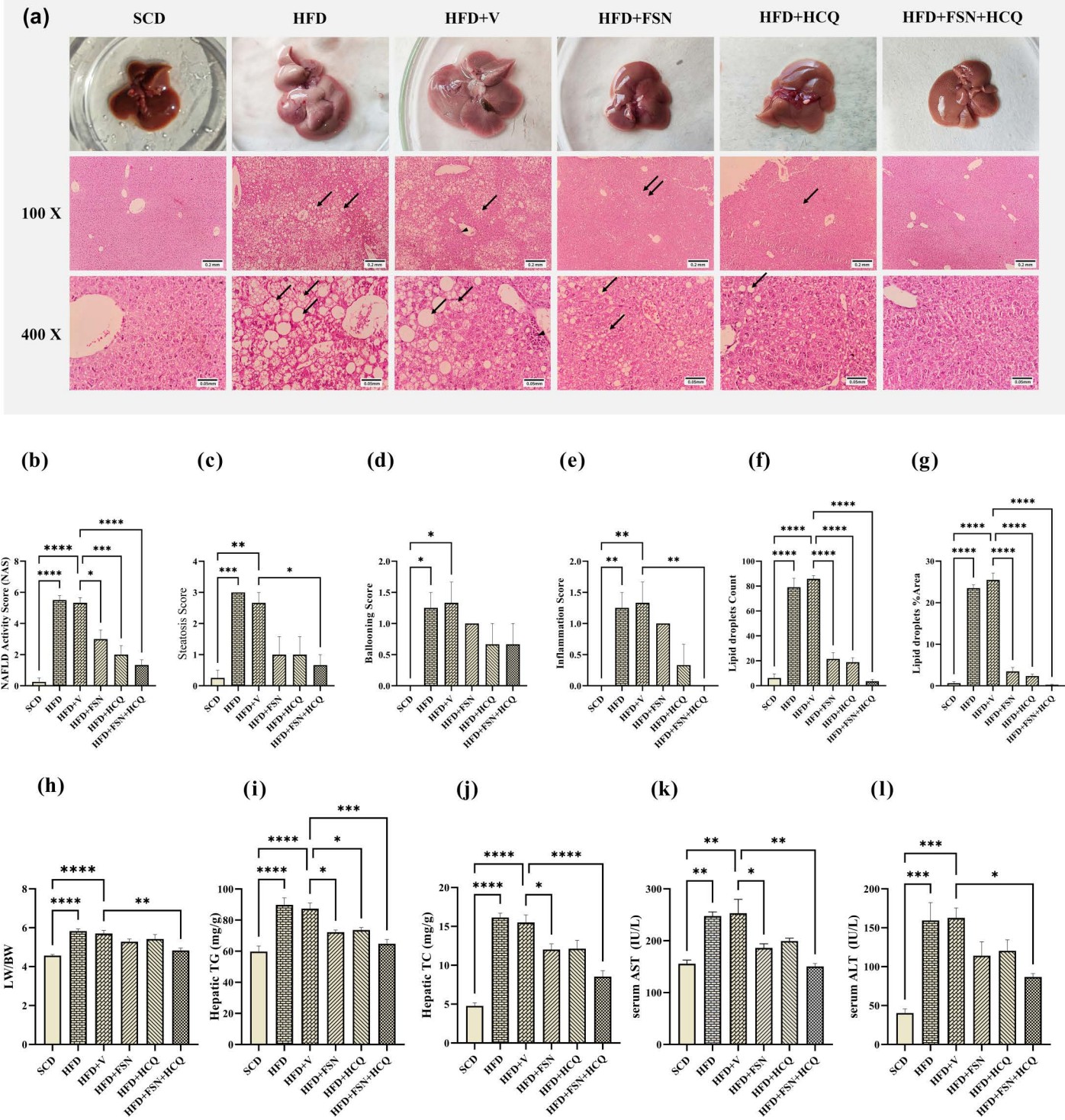

**Fig 4. Fisetin alleviates NAFLD development in HFD-fed mice. (a)** The morphology of livers (upper) and representative histological pictures of sections stained with Hematoxylin and Eosin. in the middle (100 × magnification) and the lower (400 × magnification). Small and large fat vacuoles are abundantly visible in hepatocytes in the HFD and vehicle groups (arrows). In the vehicle group, accumulation of lymphoid cells is shown (arrowhead). In the FSN and HCQ groups, fat vacuoles can be seen in some hepatocytes, but their reduction is visible. The FSN + HCQ group is very similar to the SCD group. Only a few numbers of small lipid droplets are observed in some cells. **(b)** The NAFLD activity, **(c)** steatosis, **(d)** ballooning and **(e)** lobular inflammation scores. **(f)** The lipid droplet counts and **(g)** the lipid droplet surface area percentages. **(h)** The ratio of liver weight per body weight (LW/

BW) (g/g*100). Hepatic tissue **(i)** TG and **(j)** TC content. Hepatic biochemical parameters in serum; **(k)** Aspartate aminotransferase (AST) and **(l)** Alanine aminotransferase (ALT) (n = 5). The information is displayed as mean ± S.E.M. *P < 0.05, **P < 0.01, ***P < 0.001, and ****P < 0.0001.

### Fisetin inhibits ER stress in liver of HFD-fed mice

HFD stimulates ER stress as evidenced by the significant increases in the mRNA levels of GRP78, eIF2A, ATF4, and CHOP. These effects were significantly attenuated upon administration of FSN and FSN + HCQ (Fig 5a–5d). These results demonstrated how FSN and FSN + HCQ suppress ER stress, which in turn improves the development of NAFLD.

### Fisetin induces autophagy, which lowers the accumulation of hepatic lipids in HFD-fed mice

The process by which LC3-I becomes LC3-II by conjugating with phosphatidylethanolamine (PE) is the gold standard marker for autophagosome formation [30]. Since p62 is an autophagy substrate, it is also essential. In order to assess the relationship between FSN and autophagy flux, we used western blot analysis of these variables. Western blot revealed that the levels of LC3II/LC3I ratio were significantly diminished while p62 was accumulated in liver tissue lysates of C57BL/6J mice fed on HFD (Fig 6a, 6b). After FSN treatment, LC3II/LC3I was more prominent, while p62 expression was reduced, suggesting that FSN could trigger autophagy. However, either enhanced synthesis or decreased degradation could be the cause of elevated LC3II/LC3I levels. Therefore, it may be necessary to assess autophagic flux using the lysosomal inhibitor HCQ. When HFD mice were administered HCQ, there was a slight increase in the LC3II/LC3I ratio. The combined administration of FSN along with HCQ in the sixth group led to a significant enhancement of this effect, suggesting the induction of autophagy up to the final phase and its subsequent cessation (Fig 6a, 6b). But about p62, it didn't accumulate simultaneously with a decrease in autophagy flux in the two last groups.

In order to assess autophagic activity, we also measured the levels of AMPK phosphorylation and mTOR gene expression. p-AMPK promotes autophagy by inhibiting mTOR, which in turn inhibits autophagy when activated. The expression of p-AMPK was significantly downregulated in liver of HFD-fed mice (Fig 6c). This was followed by an increased transcriptional level of mTORc-1 (Fig 6d), as inhibitor of autophagic agents, as well as decreased levels of Beclin-1 (Fig 6e) which is essential for carrying out autophagy, along with decreased levels of SQSTM1/p62, ULK1 and ATG5 (Fig 6f–6h). These alterations notably prevented after treatment with FSN, HCQ and FSN + HCQ for 8 weeks.

Altogether, our study unveils certain aspects of how FSN functions in NAFLD, as illustrated in Fig 7.

## Discussion

In recent years, several plant compounds have demonstrated potential in ameliorating- NAFLD. For instance, luteolin, resveratrol and quercetin has been shown to attenuate NAFLD [31,32]. Research conducted on animals has also reported the therapeutic potential of FSN in NAFLD. When male C57BL/6 mice were given an HFD, administering FSN orally decreased hepatic inflammation and lipid accumulation caused by the HFD by inhibiting the activation of TNF-α/RIPK3 axis [33]. In NAFLD rat model induced by an HFD, oral administration of FSN improved HNF4-α/lipin-1 signaling, lowered oxidative stress, and suppressed reactive oxygen species-induced TXNIP induction and PARP-1 activation [34].

It has been reported that the regulatory actions of FSN in mediating ER stress regulate the advancement of chronic illness [35]. Additionally, studies have shown that inducing autophagy can modulate ER stress [36]. It has not yet been thoroughly explored, though, whether FSN can control these pathways—particularly their possible interplay—in order to successfully treat NAFLD. Thus, using a mouse model of HFD-induced NAFLD, we set out to examine any potential ameliorative effects of the bioactive flavonol FSN.

In the context of this study, providing an HFD for 16 weeks allowed for the successful establishment of a C57BL/6J model of NAFLD. To elucidate the underlying molecular mechanisms, we tested the effects of FSN supplementation on

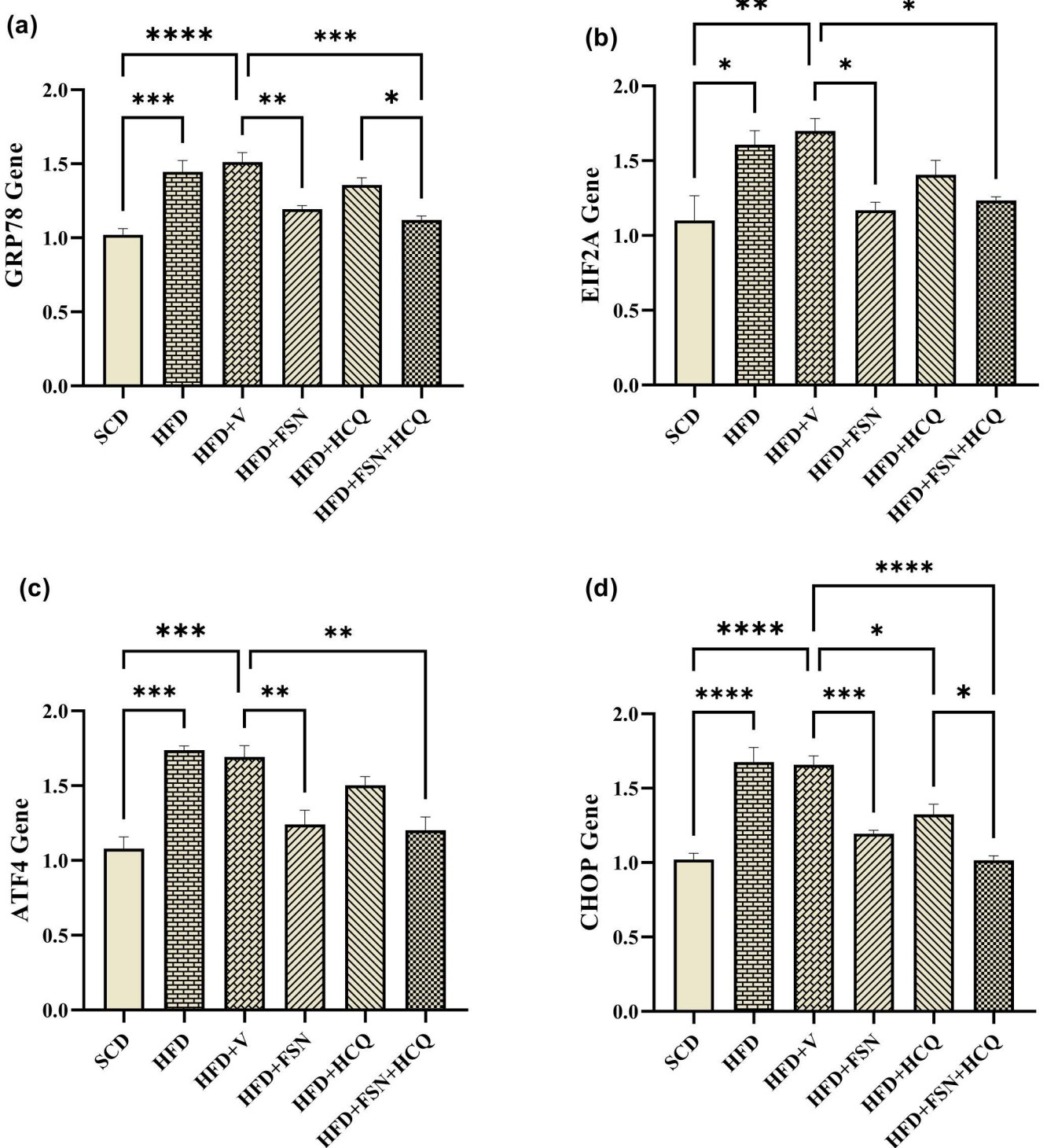

**Fig 5. Fisetin inhibits ER stress in liver of HFD-fed mice.** The relative mRNA level of **(a)** GRP78, **(b)** eIF2A, **(c)** ATF4 and **(d)** CHOP genes was quantified using real-time PCR (n = 5). The information is displayed as mean ± S.E.M. *P < 0.05, **P < 0.01, ***P < 0.001, and ****P < 0.0001.

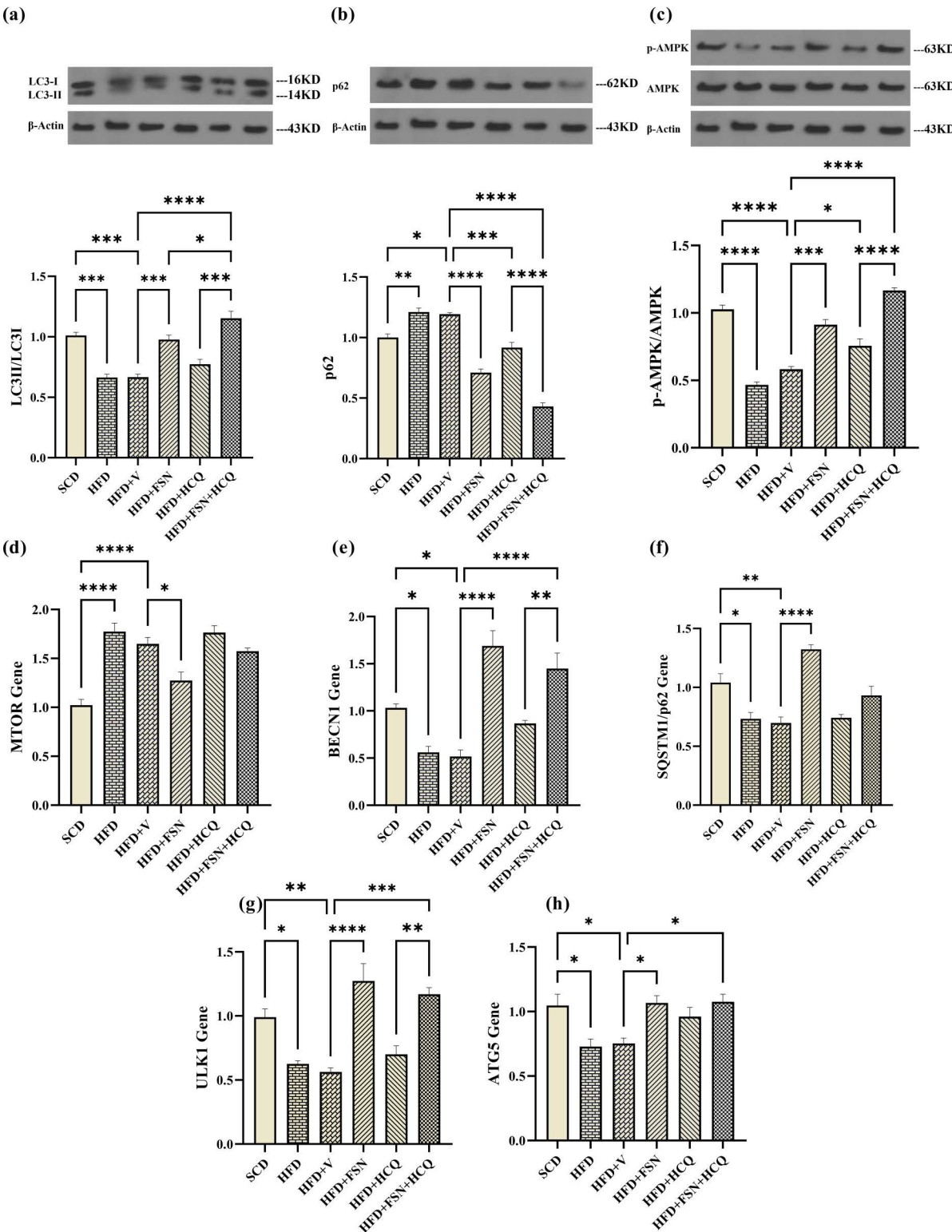

**Fig 6. Fisetin induces autophagy, which lowers the accumulation of hepatic lipids in HFD-fed mice.** The western blot results for proteins involved in autophagy; **(a)** LC3II/LC3I, **(b)** p62, and **(c)** AMPK phosphorylation in mice (n=3). The expression levels of autophagy-related genes; **(d)** MTOR, **(e)** BECN1, **(f)** SQSTM1/p62, **(g)** ULK1, and **(h)** ATG5 (n=5). The information is displayed as mean±S.E.M. *P<0.05, **P<0.01, ***P<0.001, and ****P<0.0001.

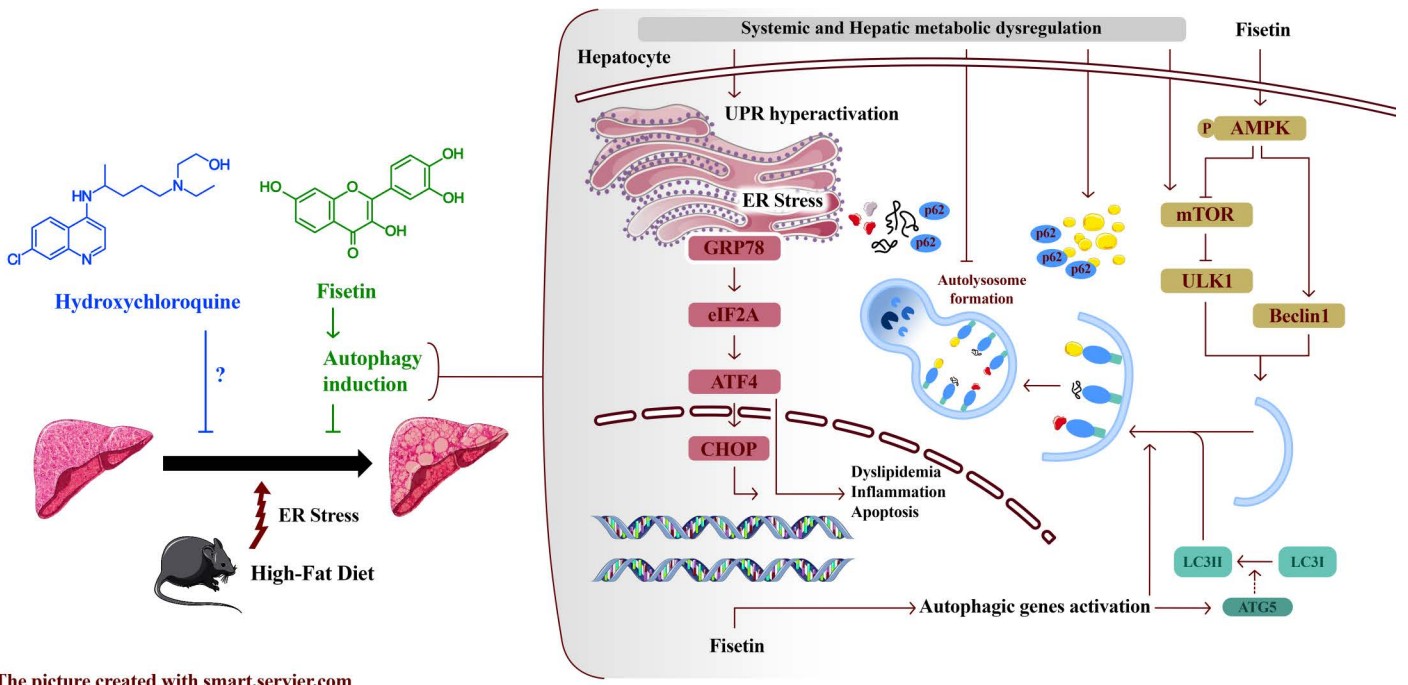

**The picture created with smart.servier.com**

**Fig 7. Fisetin functions in NAFLD.** This schematic demonstrates that HFD-induced metabolic syndrome results in heightened ER stress severity, leading to hepatic lipid accumulation, while autophagy is impaired and insufficient under these conditions. Fisetin enhances autophagic flux through the AMPK/mTOR pathway, thereby alleviating ER stress and reducing lipid accumulation.

hepatic steatosis, along with ER stress and autophagy alterations. We also used HCQ as an autophagic inhibitor, which is an immune system modulator, in combination with FSN and individually.

Here, we present novel data demonstrating that eight weeks of FSN treatment in HFD-fed mice not only suppressed lipid accumulation in hepatic tissue, but also improved the underlying pathological features of NAFLD including modulating ER stress through autophagy activation. FSN therefore has the ability to improve NAFLD brought on by obesity. Unexpectedly, an important finding of this experiment was that HCQ coadministration could reinforce the beneficial effects of FSN on hepatic lipid accumulation which may offer fresh perspectives on medical care of metabolic complications.

In the present study, we observed that eight weeks of treatment with FSN or FSN + HCQ decreased the mice weight gain, by reducing the food efficiency ratio without changing the caloric intake. In line with our results, in various studies, FSN administration has prevented body weight gain [34,37]. The HCQ group suffered from cachexia, which was in part the result of insufficient food intake. It has been shown that inhibiting mitophagy as a part of the general process of autophagy prevents HFD-induced weight gain by inhibiting intestinal lipid absorption [38]. Also, it has been seen in Fan et al.'s study that chronic treatment with CQ can cause muscle atrophy in mice [39].

In this study, treatment by FSN and more effectively by FSN + HCQ improved serum lipid profile and hepatic fat content. Several other studies showed the same results as our study regarding the effect of FSN on circulating and hepatic lipids [34,40].

It has been demonstrated that ER stress inducers can cause lipid accumulation in hepatocytes [41,42], suggesting that ER stress mediates liver steatosis [43]. HFD can also throw off the protein homeostasis of ER equilibrium, which can trigger the UPR and ER stress development [44]. According to some studies, FFAs induced ER stress in hepatocytes, as shown by the ER stress proteins, resulting in the lipids build-up and lipotoxicity [45,46]. In a different study, it was demonstrated that HFD induced ER stress in the mice livers based on the significantly higher expression of GRP78 and CHOP

[47]. Furthermore, we demonstrated that the HFD group exhibited increased mRNA expression of the UPR pathway elements, such as GRP78, elf2a, ATF4, and CHOP, in contrast to the SCD group. Multiple intracellular stress pathways may be activated as a result of ER stress. These paths can induce or hasten the development of dyslipidemia, insulin resistance (IR), inflammation, and apoptosis—all of which are involved in the pathophysiology of NAFLD [48,49]. Here, we discovered that the mice livers given FSN and FSN + HCQ had lower levels of GRP78 and CHOP gene expression than the controls. In particular, we observed reduced mRNA expression of the transcription factors elf2 and ATF4, which suggests suppression of the PERK-elf2a-ATF4 arm in the liver. This pathway holds significant importance as it controls steatosis and lipogenesis by upregulating FAS, ACC, and SCD-1 protein expression levels as well as the mRNA expression levels of SREBP-1c and ChREBP [50]. Dai et al., consistently showed that FSN mainly contributed to GRP78-mediated ER stress inhibition in an HFD-induced murine model with NAFLD and hepatocytes, which inhibited reactive oxygen species generation and mitochondrial dysfunction [47]. These results suggested the ER stress-suppressive properties of FSN, which in turn help prevent the development of NAFLD.

There are two main ways to control lipid droplet catabolism in the liver: lipolysis and lipophagy. Lipid droplets are specifically taken up by the autophagosome and transferred to the lysosome where lipase breaks them down during the lipophagy process [51]. The impairment of this is intimately related with the development of NAFLD to non-alcoholic steatohepatitis (NASH) [52]. In this investigation, the genes and proteins implicated in the three primary stages of autophagy—initiation (AMPK, mTORC1, and ULK1), nucleation (Beclin1), and elongation/maturation (Atg5 and p62 proteins, as well as the LC3BII/I ratio)—were assessed in order to assess the role of autophagy in the impact of FSN on liver steatosis.

Negative regulation of mTORC1 is one of the ways that activated p-AMPK has been demonstrated to promote autophagy [53]. Since mTORC1 signaling is the primary gate regulating autophagy, autophagy is triggered when mTORC1 is inhibited by stressors or starvation [54]. We found that FSN and FSN + HCQ treatment enhanced AMPK phosphorylation and decreased mTOR mRNA expression in compare to control. In line with these results, Liou et al showed in a study that FSN treatment increased the expression of p-AMPK in the liver of C57BL/6 male mice compared to the HFD group [37]. Jung et al indicated that FSN inhibits mTORC1 signaling via suppression of Akt activation in preadipocytes [55]. In this study, FSN group showed increments mRNA levels of genes involved in autophagy including ULK1, ATG5, and Beclin1 which can be due to mTOR inactivation. Interestingly, some studies have suggested that FSN inhibits mTORc1 activity through direct binding [56]. Collectively, our data showed that FSN applies its effects by inducing AMPK/mTOR signaling and its downstream mediators.

FSN upregulated the expression of LC3B at the protein level, along with an increase of LC3II/LC3I in liver tissue of mice. In addition, hepatic p62 protein levels lowered in the FSN-treated mice, indicating activation of autophagic flux by this natural product. Ding et al showed that FSN promotes mitophagy through upregulating LC3-II and downregulating p62 in cardiac microvascular endothelial cells (CMECs) of rats with sepsis-associated encephalopathy [57]. Moreover, in a study by Zhang et al, the administration of FSN (80 mg/kg) provoked LC3 puncta accumulation and promoted the expression of p62 in acetaminophen-damaged liver tissue of Mice [58]. Our study found that autophagy was inhibited in liver of HFD-mice, leading to increased lipid accumulation in the liver. While, FSN effectively reversed these alterations. These findings suggest that FSN improves intracellular lipid metabolism, which is at least in part result from the promotion of autophagic flux.

A more interesting finding in our study is that the decrease in p62 is despite the increase in its gene transcription induced by FSN. It seems that a part of a larger decrease in p62 is compensated and so covered by the increase in its production. Therefore, the net result is only a slight decrease in p62 protein, while we observed an increase in autophagy activity in this group by the LC3II/LC3I ratio as an autophagy hallmark. Kim et al have also shown an increase in relative mRNA level of SQSTM1/p62 expression with FSN in T4 cells and neurons [59]. The concentration of this protein is actually subject to intricate modifications at the transcription and post-transcriptional levels. As a result, the measurement of p62 content as a proxy for autophagy activity remains debatable and is largely susceptible to interpretation errors.

This matter has also been seen in the case of other autophagy inducers such as resveratrol, which exert their effect on both levels of protein synthesis and degradation [60]. In order to prevent misunderstandings, it could be instructive to employ pharmaceuticals that block either the input or output current. Unexpectedly, HCQ co-treatment did not cause an increase in protein levels, whereas inhibition of autophagy is expected to be associated with protein accumulation as seen in studies performed on cell lines [61]. To the interpretation of this result, it is possible to refer to two facts. Firstly, in the case of SQSTM1/p62, several transcription factors, including AP-1, NF-κB, Ets-1, and SP-1, have binding sites located in the 5'-flanking region of its promoter [62], as well as, its protein's structure is made up of several important domains that allow it to interact with crucial elements in vital signaling pathways. [60]. Hence, p62 serves as both an autophagic cargo receptor and a major signaling hub that connects a number of crucial pro- and anti-inflammatory pathways in this instance [63]. Secondly, HCQ is not just an autophagy inhibitor. It has been emphasized that the interpretation of the results of blocking autophagy requires caution because HCQ and CQ can cause multiple cellular changes [64]. In this context, the results of the present study showed that HCQ therapy is accompanied by a decrease in SQSTM1/p62 gene expression. It is possible that since a prolonged treatment with HCQ was used in this study rather than cell studies, the contribution of the transcription downregulation in the cellular p62 pool could be more pronounced than the accumulation due to an acute inhibition of autophagy. Here, the reduced SQSTM1/p62 transcription could be in response to the general inflammation suppression by HCQ. Therefore, it may be better to consider SQSTM1/p62 transcriptional activity along with its protein expression to determine autophagy activity. For example, a ratio of these two could be more efficient and correct, although more studies are needed.

ER stress can trigger autophagy in mammalian cells, helping to remove misfolded proteins and damaged ER components, reducing ER burden, and promoting cell survival [65,66]. This study also demonstrated a connection between elevated ER stress and decreased autophagic flux in C57BL/6/J fed on HFD. Lastly, to bolster our theory that FSN-induced enhanced autophagy can impede the advancement of NAFLD caused by ER stress, Feng et al. have shown that GRP78 upregulates autophagy via AMPK phosphorylation and mTOR inhibition. In contrast, the autophagy inducer rapamycin alleviates ER stress to support cell survival [67]. Consistently, our study confirmed a direct link between p-AMPK levels and the inactivation of mTOR gene transcription, leading to a decrease in GRP78 expression.

The findings of our study have raised one query: why, in spite of autophagy inhibition, does the FSN+HCQ group exhibit greater improvement than the FSN group? To address this question, further research is needed, particularly to examine the synergistic implications of FSN and HCQ on metabolic dysfunction. Interpreting the data in animal research is more complicated than in cell culture investigation to ascertain the part of any underlying mechanism. Still, it also gives us a more comprehensive and insightful perspective. In light of the aforementioned, it should be considered that ER dysfunction and immunometabolism signaling are integrated into multiple vicious cycles [68]. The potential mechanisms through which HCQ may assist FSN in improving NAFLD include several factors. Beyond HCQ's anti-inflammatory effects [69], it appears to have a specific impact on lipid metabolism, as noted by Qiao et al., who reported that HCQ reduces hepatic lipogenesis by decreasing the expression of FAS and ACC [70]. Additionally, in another study where CQ was administered in combination with sulforaphane to HFD-fed mice, the favorable effects of sulforaphane on metabolic dysfunction via inhibition of hepatic lipogenesis were enhanced by CQ [71]. These reports align with our study's findings regarding improved serum lipid profiles and NAS in the HCQ group, which were somewhat similar to those observed with FSN. Another possible factor we must consider from our evidence is that the HCQ+FSN group consumed relatively less food compared to other HFD groups (although this difference was statistically insignificant), which might be another reason for the lipid profile regulation and NAFLD improvement.

Our study has shown that HCQ also plays a role in regulating glucose and insulin homeostasis, a finding that has been confirmed by several other studies. It is demonstrated that HCQ improves insulin sensitivity by reducing PPARγ levels and stimulating glucose uptake. Moreover, it significantly increases the phosphorylation of Irs1 and Akt in liver tissue [70]. This drug may have multifaceted effects on glucose regulation, including improved insulin sensitivity, increased

insulin secretion, reduced hepatic insulin clearance, decreased intracellular insulin degradation, and favorable impacts on adipocytokines [72]. In diabetic patients, HCQ has also shown significant improvements in insulin levels, HbA1c, FPG, and postprandial blood glucose levels [73]. Regarding the similar effects observed between the HFD + HCQ and HFD + FSN + HCQ groups in glucose homeostasis, it is possible that FSN and HCQ share overlapping biochemical targets and similar mechanisms for glucose-lowering. Therefore, their simultaneous use may not result in a significant synergistic effect on glucose levels. Thus, the greater improvement in NAFLD in the HFD + FSN + HCQ group may be more related to other factors, such as inflammation, rather than a direct effect on blood glucose.

In this study, we acknowledge several limitations that should be addressed in future investigations. First, the assessment of mTORC1 and ULK1 activity was limited to transcriptional analysis. To gain a more comprehensive understanding of their roles in autophagy regulation, future studies should evaluate the phosphorylation states of these proteins. Additionally, while we provided insights into the mRNA levels of ER stress markers, we recognize the limitation of not measuring protein levels or their phosphorylated states. Incorporating these analyses would enhance our understanding of the impact of ER stress on cellular functions. Furthermore, the current study demonstrated a reduction in ER stress markers with FSN treatment; however, employing ER stress inducers in conjunction with FSN treatment would provide stronger confirmation of whether FSN's effects on autophagy and lipid accumulation are indeed mediated through ER stress modulation. Lastly, we recognize the importance of investigating autophagy and ER stress in isolated hepatocytes treated with FSN or in combination with HCQ to ensure that the observed effects are directly attributable to hepatocytes during NAFLD treatment. We will certainly consider them for future studies to complement our current findings.

In summary, we presented new research demonstrating that HFD-induced autophagic impairment was associated with hepatic lipid accumulation and ER stress burden, while FSN- treated group had an enhanced autophagic flux along with lowered lipid accumulation and ER stress intensity. Hence, these results suggest that FSN treatment prevented the lipid accumulation in HFD-fed mice, by restoring hepatic autophagy and subsequently reducing the ER stress.

## Commitment to research integrity

We hereby confirm that the study has been reported in accordance with the ARRIVE guidelines, ensuring transparency and completeness in the reporting of our research methods, results, and conclusions.

## Supporting information

**S1 Table. Primers sequences used in this study.**
(DOCX)

## Acknowledgments

We would like to extend our appreciation to Tehran Daru Pharmaceutical Company for their generous provision of hydroxychloroquine sulfate for our research project.

We would like to acknowledge the assistance of an AI writing assistant in improving the clarity and quality of the text presented in this article.

## Author contributions

**Conceptualization:** Reza Meshkani, Ghodratollah Panahi.

**Data curation:** Mahboobe Sattari, Sadra Samavarchi Tehrani.

**Formal analysis:** Golnaz Goodarzi.

**Funding acquisition:** Ozra Tabatabaei-Malazy, Ghodratollah Panahi.

**Investigation:** Maryam Akhavan Taheri.

**Methodology:** Ghodratollah Panahi.

**Project administration:** Mahboobe Sattari.

**Software:** Sadra Samavarchi Tehrani.

**Supervision:** Mohammad Esmaeil Shahaboddin, Ghodratollah Panahi.

**Validation:** Ehsan Khalili.

**Visualization:** Maryam Akhavan Taheri.

**Writing – original draft:** Mahboobe Sattari.

**Writing – review & editing:** Ghodratollah Panahi.

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
