## [Decision Letter · Decision Letter 0]

18 Nov 2024

PONE-D-24-31421Therapeutic potential of fisetin in hepatic steatosis: insights into autophagy pathway regulation and endoplasmic reticulum stress alleviation in high-fat diet-fed mice

Dear Dr. Panahi,

Thank you for submitting your manuscript to PLOS ONE. After careful consideration, we feel that it has merit but does not fully meet PLOS ONE’s publication criteria as it currently stands. Therefore, we invite you to submit a revised version of the manuscript that addresses the points raised during the review process.

Additional experiements are required in order to conclude that Fisetin has indeed a role on liver and that ER stress and autophagy are really modulated

We look forward to receiving your revised manuscript.

Kind regards,

Catherine Mounier

Academic Editor

PLOS ONE

Journal Requirements:

2. To comply with PLOS ONE submissions requirements, in your Methods section, please provide additional information regarding the experiments involving animals and ensure you have included details on methods of anesthesia and/or analgesia, and efforts to alleviate suffering.

5. In the online submission form you indicate that your data is not available for proprietary reasons and have provided a contact point for accessing this data. Please note that your current contact point is a co-author on this manuscript. According to our Data Policy, the contact point must not be an author on the manuscript and must be an institutional contact, ideally not an individual. Please revise your data statement to a non-author institutional point of contact, such as a data access or ethics committee, and send this to us via return email. Please also include contact information for the third party organization, and please include the full citation of where the data can be found.

Reviewers' comments:

Reviewer's Responses to Questions

**Comments to the Author**

1. Is the manuscript technically sound, and do the data support the conclusions?

Reviewer #1: Yes

Reviewer #2: Partly

2. Has the statistical analysis been performed appropriately and rigorously? 

Reviewer #1: Yes

Reviewer #2: Yes

3. Have the authors made all data underlying the findings in their manuscript fully available?

Reviewer #1: Yes

Reviewer #2: Yes

4. Is the manuscript presented in an intelligible fashion and written in standard English?

Reviewer #1: Yes

Reviewer #2: Yes

5. Review Comments to the Author

Reviewer #1: Reviewer comments:

The manuscript titles “Therapeutic potential of fisetin in hepatic steatosis: insights into autophagy pathway regulation and endoplasmic reticulum stress alleviation in high-fat diet-fed mice” by Sattari et al demonstrates the efficacy of Fisetin in the amelioration of NAFLD through alleviation of ER stress and enhanced autophagy. The authors show in mice model of HFD, that administration of Fisetin targeted autophagy and ER stress pathway and increased AMPH phosphorylation and helped in the management of NAFLD. Although the study is of interest in the current scenario of metabolic disease over-burden, following comments should be addressed to make the study suitable for publication.

Major Comments:

1. Can the authors explain the similar effects seen between HFD+FSN and HFD+HCQ for glucose and insulin homeostasis in Fig 2.

2. No difference in glucose homeostasis is observed for HFD+HCQ and HFD+FNS+HCQ groups in the physiological assays in Fig 2 and 3.

3. Since the study is based on role of Fisetin on autophagy in NAFLD amelioration, can the authors explain the lack of any deleterious seen for HFD+HCQ group, since HCQ is known to inhibit autophagic process completely. As reported by the authors in Fig 4, NAFLD, steatosis, ballooning and inflammation scores are either similar or lower than HFD+FSN group.

4. Authors are requested to compare the statistical significance of the data between HFD+HCQ and HFD+FNS+HCQ groups, wherever applicable.

5. There are some regularly histology experiments that are missing from the study as listed below:

a) ORO stain in liver sections

b) Sirius red stain

c) a-SMA staining

6. Quantification for fat vacuoles for Figure 4.

7. The authors requested to clarify the model in Fig 7, where they have shown Fisetin to inhibit autophagy, similar to HCQ.

8. The authors should do experiments related to autophagy and ER stress analysis in isolated hepatocytes treated Fisetin or in combination with HCQ to demonstrate the direct effect of Fisetin on authophagy in hepatocytes. This would ensure that the effect of Fisetin is directly on hepatocytes during NAFLD treatment and not through other systemic effects.

Minor Comments:

1. References should be provided for Lines 81-83, page 3.

2. Current guidelines for Glucose tolerance tests advise a fasting period of 6hrs before the experiment. The authors are requested to explain why they have selected 10hrs. The authors should also clearly mention the time the GTT and ITT experiments were done.

3. the authors are requested to mention the housekeeping gene used for qPCR normalization and indicate whether it showed any differences between the treated groups.

4. Please indicate whether totalAUC or incremental AUC was used to plot area under the curve for GTT and ITT experiments.

5. Please indicate individual data points in the graphs to indicate numer of samples and the sread of the data. Also include the exact mice numbers used for each experiment and the criteria on which any mouse was not included.

Reviewer #2: In this manuscript, Sattari and co-authors investigated fisetin's influence on ER stress, autophagy pathway regulation, and hepatic lipid accumulation, providing promising insights into its potential for mitigating liver dysfunction. While the study is well-conceived and provides valuable insights into FSN’s potential in NAFLD, several issues need to be addressed to strengthen the manuscript.

Major Points.

1. The manuscript demonstrates a reduction in ER stress markers with FSN treatment. To strengthen this argument an experiment utilizing ER stress inducers (e.g., tunicamycin) in conjunction with FSN treatment could confirm whether FSN’s effects on autophagy and lipid accumulation are indeed mediated through ER stress modulation.

2. Figure 6a: The quantitative graphics show results that differ from those observed in the western blot images. I suggest the authors replace the images.

3. Figure 6d: The transcriptional level of mTORC1 alone is not sufficient to indicate its activity. To accurately assess mTORC1 activity, it is necessary to analyze its phosphorylation state or the phosphorylation of its downstream target, S6K.

4. Figure 6g: The transcriptional level of ULK1 is not sufficient to indicate autophagic activity. The most suitable indicator for assessing autophagy activation is the phosphorylation of ULK1 at activating sites (e.g., Ser317 or Ser555). ULK1 phosphorylation at these sites is closely associated with the direct activation of the autophagy pathway, making it a key indicator of whether autophagy has been initiated.

5. Figure 5: While mRNA levels of ER stress markers provide valuable insights, they should be complemented with protein-level and functional analyses to accurately assess ER stress and its cellular impact. To comprehensively evaluate ER stress, The authors should measure protein levels of key ER stress markers (e.g., GRP78, CHOP) or their phosphorylated states (e.g., p-PERK, p-eIF2α).

6. The manuscript suggests that FSN and HCQ together are more effective but does not explain the mechanism behind this synergy. the authors should discuss synergistic effects of FSN and HCQ.

Minor Points.

1. Correct the typo in Figure 4e’s y-axis title from "scor" to "score"

6. PLOS authors have the option to publish the peer review history of their article (what does this mean? ). If published, this will include your full peer review and any attached files.

**Do you want your identity to be public for this peer review?** For information about this choice, including consent withdrawal, please see our Privacy Policy .

Reviewer #1: No

Reviewer #2: **Yes: ** Hwan-Woo Park

---

## [Author Response · Author response to Decision Letter 1]

5 Mar 2025

January 25,2025

SUBJECT: Manuscript Revision

Manuscript Number: PONE-D-24-31421

Dear Reviewers / Editorial Team,

We would like to express our sincere gratitude for the time and effort you dedicated to reviewing our manuscript. Your constructive comments have been invaluable in enhancing the quality of the work. We truly appreciate your insightful feedback, which will undoubtedly contribute to the success of the article. We understand the importance of having a well-referenced article and have made every effort to enhance this aspect accordingly.

In the following sections, I will address each of the points raised in your comments in detail.

Reviewer #1: Reviewer comments:

The manuscript titles “Therapeutic potential of fisetin in hepatic steatosis: insights into autophagy pathway regulation and endoplasmic reticulum stress alleviation in high-fat diet-fed mice” by Sattari et al demonstrates the efficacy of Fisetin in the amelioration of NAFLD through alleviation of ER stress and enhanced autophagy. The authors show in mice model of HFD, that administration of Fisetin targeted autophagy and ER stress pathway and increased AMPK phosphorylation and helped in the management of NAFLD. Although the study is of interest in the current scenario of metabolic disease over-burden, following comments should be addressed to make the study suitable for publication.

Major Comments:

1. Can the authors explain the similar effects seen between HFD+FSN and HFD+HCQ for glucose and insulin homeostasis in Fig 2.

2. No difference in glucose homeostasis is observed for HFD+HCQ and HFD+FNS+HCQ groups in the physiological assays in Fig 2 and 3.

Thank you for your insightful comments (1,2) regarding the glucose and insulin homeostasis observations. We have addressed your concerns by discussing the effects of HCQ on glucose metabolism and the relationship between HFD+HCQ and HFD+FSN+HCQ groups in the discussion section on page 16-17, line 466-479.

3. Since the study is based on role of Fisetin on autophagy in NAFLD amelioration, can the authors explain the lack of any deleterious seen for HFD+HCQ group, since HCQ is known to inhibit autophagic process completely. As reported by the authors in Fig 4, NAFLD, steatosis, ballooning and inflammation scores are either similar or lower than HFD+FSN group.

Thank you for your insightful comments regarding the lack of deleterious effects of HCQ in our study. We have addressed your concerns by discussing the mechanisms through which HCQ may interact with FSN to improve metabolic outcomes in the context of NAFLD. This discussion has been included on page 16, line 454-465 of the manuscript.

4. Authors are requested to compare the statistical significance of the data between HFD+HCQ and HFD+FNS+HCQ groups, wherever applicable.

Thank you for your suggestion. We have conducted statistical comparisons between the HFD+HCQ and HFD+FNS+HCQ groups for all relevant variables. The following variables showed significant differences: GRP78, CHOP, ULK1, p62, LC3II/LC3I, p-AMPK/AMPK, BECN1, and Food Efficiency Ratio. We have indicated these significant differences in the graphs.

5. There are some regularly histology experiments that are missing from the study as listed below:

a) ORO stain in liver sections

b) Sirius red stain

c) a-SMA staining

Thank you for your insightful comments. We appreciate your feedback regarding the histological experiments. While we acknowledge that ORO staining is more specific for lipid droplet detection, we believe that the H & E staining we performed can still effectively demonstrate the presence of lipid vacuoles in liver sections. In fact, we are planning to refine our analysis based on another comment provided by you.

Regarding the other two stains, Sirius red stain is typically used for collagen detection, and α-SMA staining assesses the degree of fibrosis. However, in our study, we based our approach on prior experience with a model that focuses on fatty liver in the steatosis stage of NAFLD. The high-fat diet during this timeframe does not induce the fibrosis seen in NASH. Therefore, we believe that incorporating these additional stains may not significantly enhance our findings. We appreciate your understanding and look forward to your further input.

6. Quantification for fat vacuoles for Figure 4.

Thank you for your valuable comment regarding the quantification of fat vacuoles for Figure 4. We have performed the quantification as outlined in the methods section of the article (page 6, line 160-162). The analysis was carried out using ImageJ software, focusing on two variables: particle count and percentage of their surface area. We have also included two new graphs in Figure 4 (Figs 4f,g) that illustrates them.

7. The authors requested to clarify the model in Fig 7, where they have shown Fisetin to inhibit autophagy, similar to HCQ.

Thank you for your insightful comment. We acknowledge that this part of the figure may appear ambiguous and could lead to misinterpretation. The two inhibitory symbols were meant to indicate the prevention of NAFLD by the treatments, specifically FSN and HCQ. FSN exerts this effect through the induction of autophagy, while the mechanism of action for HCQ remains unclear. We have revised this section of the figure to accurately convey our intended message without confusion.

8. The authors should do experiments related to autophagy and ER stress analysis in isolated hepatocytes treated Fisetin or in combination with HCQ to demonstrate the direct effect of Fisetin on autophagy in hepatocytes. This would ensure that the effect of Fisetin is directly on hepatocytes during NAFLD treatment and not through other systemic effects.

Thank you for your valuable feedback. We recognize the importance of further investigating autophagy and ER stress in isolated hepatocytes treated with FSN or in combination with HCQ to demonstrate the direct effect of FSN on autophagy in hepatocytes. However, we face specific constraints regarding time and available resources, compounded by the financial limitations and the current economic situation in our country due to sanctions, which prevent us from conducting the suggested experiments at this stage. Additionally, in the current study, we have explored the effects of FSN on autophagy and ER stress through in vivo experiments, allowing us to examine the complex systemic and organ-specific mechanisms involved. We have also acknowledged these constraints as a limitation in the discussion section of the manuscript (on page 17, line 480-493).

Minor Comments:

1. References should be provided for Lines 81-83, page 3.

Thank you for your feedback. We have added the appropriate references.

2. Current guidelines for Glucose tolerance tests advise a fasting period of 6hrs before the experiment. The authors are requested to explain why they have selected 10hrs. The authors should also clearly mention the time the GTT and ITT experiments were done.

Thank you for your valuable feedback regarding the fasting period for the GTT and ITT. For our experiments, we followed the guidelines outlined in the article titled "Guidelines and Considerations for Metabolic Tolerance Tests in Mice." According to these guidelines, we implemented an overnight fasting period of 10 hours (from 8 PM to 6 AM) for the GTT and a 6-hour fasting period (from 6 AM to 12 PM) for the ITT. We have also added the specific timing of these experiments to the methods section of our manuscript.

3. the authors are requested to mention the housekeeping gene used for qPCR normalization and indicate whether it showed any differences between the treated groups.

Thank you for your valuable suggestion. We used β-actin as the housekeeping gene for qPCR normalization, and we found that its expression did not show any significant differences between the treated groups. This information has been included in the Methods section of the manuscript.

4. Please indicate whether total AUC or incremental AUC was used to plot area under the curve for GTT and ITT experiments.

Thank you for your insightful comment regarding the AUC for the GTT and ITT experiments. We used the total AUC for both experiments, and we have indicated this in the manuscript as well.

5. Please indicate individual data points in the graphs to indicate number of samples and the spread of the data. Also include the exact mice numbers used for each experiment and the criteria on which any mouse was not included.

We appreciate the constructive feedback, in response to your suggestion, we would like to clarify that, as stated in the Materials and Methods section of our manuscript, each experimental group consisted of eight mice. This number was chosen to ensure adequate statistical power while adhering to ethical considerations regarding animal use.

Reviewer #2: In this manuscript, Sattari and co-authors investigated fisetin's influence on ER stress, autophagy pathway regulation, and hepatic lipid accumulation, providing promising insights into its potential for mitigating liver dysfunction. While the study is well-conceived and provides valuable insights into FSN’s potential in NAFLD, several issues need to be addressed to strengthen the manuscript.

Major Points.

1. The manuscript demonstrates a reduction in ER stress markers with FSN treatment. To strengthen this argument an experiment utilizing ER stress inducers (e.g., tunicamycin) in conjunction with FSN treatment could confirm whether FSN’s effects on autophagy and lipid accumulation are indeed mediated through ER stress modulation.

Thank you for your valuable suggestion. You are correct that employing ER stress inducers like tunicamycin in conjunction with FSN treatment would provide a stronger confirmation of whether FSN’s effects on autophagy and lipid accumulation are mediated through ER stress modulation. Our primary aim in this study was to investigate the effects of FSN on autophagy, which is why we specifically used an autophagy inhibitor. Unfortunately, due to budget limitations, we were unable to expand the study design to include additional experiments. However, we appreciate your insight and will certainly consider this approach for future studies that could complement our current findings.

2. Figure 6a: The quantitative graphics show results that differ from those observed in the western blot images. I suggest the authors replace the images.

Thank you for your comment regarding Figure 6a. The discrepancies between the quantitative graphics and the western blot images arise from the fact that the quantification is based on data from multiple gels rather than a single gel. We selected the blot that best aligns with the overall quantitative results, but variability among the gels can lead to differences that do not reflect one another perfectly, as each gel may show slight variations. Therefore, we believe that the current representation is more appropriate.

3. Figure 6d: The transcriptional level of mTORC1 alone is not sufficient to indicate its activity. To accurately assess mTORC1 activity, it is necessary to analyze its phosphorylation state or the phosphorylation of its downstream target, S6K.

4. Figure 6g: The transcriptional level of ULK1 is not sufficient to indicate autophagic activity. The most suitable indicator for assessing autophagy activation is the phosphorylation of ULK1 at activating sites (e.g., Ser317 or Ser555). ULK1 phosphorylation at these sites is closely associated with the direct activation of the autophagy pathway, making it a key indicator of whether autophagy has been initiated.

5. Figure 5: While mRNA levels of ER stress markers provide valuable insights, they should be complemented with protein-level and functional analyses to accurately assess ER stress and its cellular impact. To comprehensively evaluate ER stress, the authors should measure protein levels of key ER stress markers (e.g., GRP78, CHOP) or their phosphorylated states (e.g., p-PERK, p-eIF2α).

Thank you for your thoughtful comments (3-5) on our manuscript. You are correct that analyzing the phosphorylation state of mTORC1 and its downstream targets, as well as the phosphorylation of ULK1 at key activating sites, would provide a more accurate assessment of their activities. Additionally, while the transcriptional levels of ER stress markers are informative, complementing them with protein-level and functional analyses is essential for a comprehensive evaluation of ER stress and its cellular impact.

Our primary goal in this study was to investigate the effect of FSN on hepatic steatosis, building on previous research that highlighted its influence through metabolic and inflammatory pathways. Here, we aimed to conduct preliminary work to assess FSN's effects on hepatic steatosis via autophagy pathways. Despite our strong desire to produce a thoroughly robust and scientifically rigorous study, financial constraints and the current economic situation in our country, compounded by existing sanctions, limit our capability to conduct further tests. Therefore, we are compelled to work within these constraints. We have also acknowledged these limitations in the discussion section of the manuscript (on page 17, line 480-493). However, we greatly appreciate your suggestions and will certainly consider them for future studies that could complement this research.

6. The manuscript suggests that FSN and HCQ together are more effective but does not explain the mechanism behind this synergy. the authors should discuss synergistic effects of FSN and HCQ.

Thank you for your valuable feedback. We appreciate your suggestion to discuss the synergistic effects of FSN and HCQ. In response, we have added potential mechanisms through which HCQ may enhance the effects of FSN in the treatment of NAFLD in the discussion section on page 16, line 454-465.

Minor Points.

1. Correct the typo in Figure 4e’s y-axis title from "scor" to "score"

Thank you for pointing out the typo in Figure title. We have corrected "scor" to "score".

Thank you once again for your support and guidance.

Best regards,

Ghodratollah (Shahriyar) Panahi, Assistant Professor,

Dept. Of Clinical Biochemistry, Faculty of Medicine,

Tehran University of Medical Sciences, Tehran, Iran,

pshahriyar@gmail.com

ghpanahi@sina.tums.ac.ir

---

## [Decision Letter · Decision Letter 1]

19 Mar 2025

Therapeutic potential of fisetin in hepatic steatosis: insights into autophagy pathway regulation and endoplasmic reticulum stress alleviation in high-fat diet-fed mice

PONE-D-24-31421R1

Dear Dr. Panahi,

We’re pleased to inform you that your manuscript has been judged scientifically suitable for publication and will be formally accepted for publication once it meets all outstanding technical requirements.

Kind regards,

Catherine Mounier

Academic Editor

PLOS ONE

Additional Editor Comments (optional):

Reviewers' comments:

Reviewer's Responses to Questions

**Comments to the Author**

1. If the authors have adequately addressed your comments raised in a previous round of review and you feel that this manuscript is now acceptable for publication, you may indicate that here to bypass the “Comments to the Author” section, enter your conflict of interest statement in the “Confidential to Editor” section, and submit your "Accept" recommendation.

Reviewer #2: All comments have been addressed

2. Is the manuscript technically sound, and do the data support the conclusions?

Reviewer #2: Yes

3. Has the statistical analysis been performed appropriately and rigorously? 

Reviewer #2: Yes

4. Have the authors made all data underlying the findings in their manuscript fully available?

Reviewer #2: Yes

5. Is the manuscript presented in an intelligible fashion and written in standard English?

Reviewer #2: Yes

6. Review Comments to the Author

Reviewer #2: The manuscript is scientifically sound, well-structured, and provides reasonable justifications for the limitations in experimental design. Given the rigorous methodology and clear data presentation, I recommend acceptance in its current form.

7. PLOS authors have the option to publish the peer review history of their article (what does this mean? ). If published, this will include your full peer review and any attached files.

**Do you want your identity to be public for this peer review?** For information about this choice, including consent withdrawal, please see our Privacy Policy .

Reviewer #2: **Yes: ** Hwan-Woo Park

---

## [Editor Report · Acceptance letter]

PONE-D-24-31421R1

PLOS ONE

Dear Dr. Panahi,

I'm pleased to inform you that your manuscript has been deemed suitable for publication in PLOS ONE. Congratulations! Your manuscript is now being handed over to our production team.

Kind regards,

on behalf of

Dr. Catherine Mounier

Academic Editor

PLOS ONE